# Comprehensive Bioinformatics Analysis the circRNAs of Viral Infection Associated Pathway in HepG2 Expressing ORF3 of Genotype IV Swine Hepatitis E Virus

**DOI:** 10.3390/microorganisms13122654

**Published:** 2025-11-22

**Authors:** Hanwei Jiao, Lingjie Wang, Chi Meng, Shengping Wu, Yubo Qi, Jianhua Guo, Jixiang Li, Liting Cao, Yu Zhao, Jake J. Wen, Fengyang Wang

**Affiliations:** 1The College of Veterinary Medicine, Southwest University, Chongqing 402460, China; jiaohanwei@swu.edu.cn (H.J.); guolicheng666@email.swu.edu.cn (L.W.); mengchi@email.swu.edu.cn (C.M.); chemie@email.swu.edu.cn (S.W.); qyb314159@email.swu.edu.cn (Y.Q.); guo0619@swu.edu.cn (J.G.); swu_lucky@163.com (J.L.); caoliting@swu.edu.cn (L.C.); 2Ministry of Agriculture and Rural Affairs Key Laboratory of Crop Genetic Resources and Germplasm Innovation in Karst Region, Institute of Animal Husbandry and Veterinary Medicine of Guizhou Academy of Agricultural Science, Guiyang 550005, China; zhaoyu@gzsnky.wecom.work; 3Center for Translational Cancer Research, Brown Foundation Institute of Molecular Medicine, 1825 Pressler St., Suite 310, Houston, TX 77030, USA; 4Hainan Key Laboratory of Tropical Animal Reproduction & Breeding and Epidemic Disease Research, School of Tropical Agriculture and Forestry, Hainan University, Haikou 570228, China

**Keywords:** SHE, SHEV ORF3, HepG2, viral infection, circRNA-miRNA-mRNA network

## Abstract

The open reading frame 3 (ORF3) protein of the swine hepatitis E virus (SHEV) is a critical virulence factor implicated in viral infection, yet its precise mechanisms remain poorly understood. Circular RNAs (circRNAs) have emerged as key regulators of gene expression during viral infections by functioning as miRNA sponges. This study aimed to identify key circRNAs and construct a potential circRNA-miRNA-mRNA regulatory network associated with the viral infection pathway in HepG2 cells expressing genotype IV SHEV ORF3. Based on our previous high-throughput circRNA and transcriptome sequencing data from HepG2 cells with adenovirus-mediated ORF3 overexpression, we screened for differentially expressed circRNAs and mRNAs linked to viral infection pathways. Using bioinformatic tools, we predicted miRNAs targeted by these mRNAs and those that could bind to the circRNAs, ultimately constructing a competing endogenous RNA (ceRNA) network with Cytoscape. We identified 31 differentially expressed circRNAs and 7 mRNAs (HSPA8, HSPA1B, EGR2, CXCR4, SOCS3, NOTCH3, and ZNF527) related to viral infection. A potential ceRNA network comprising 32 circRNAs, 23 miRNAs, and the 7 mRNAs was constructed. Core circRNAs, including ciRNA203, circRNA14936, and circRNA5562, may act as miRNA sponges to regulate the expression of these mRNAs. This network suggests a novel mechanism by which SHEV ORF3 might modulate host cell functions to facilitate viral infection.

## 1. Introduction

Hepatitis E Virus (HEV) is a single-stranded, positive-sense RNA virus belonging to the genus Orthohepevirus within the family Hepeviridae. With a genome of approximately 7.2 kilobases, HEV is a significant human pathogen [1]. Infection primarily leads to self-limiting acute viral hepatitis; however, it can also cause chronic hepatitis in immunocompromised individuals [2,3]. Currently, HEV comprises eight distinct genotypes (HEV1–8) and 36 subtypes. HEV1 and HEV2 are typically restricted to human-to-human transmission and are endemic in developing regions of Africa and Asia, manifesting as sporadic or localized outbreaks [4]. In contrast, HEV3 and HEV4 are recognized as zoonotic pathogens, frequently isolated from both humans and animals, suggesting their role in cross-species transmission [5]. Regarding swine hepatitis E virus (SHEV), four genotypes have been identified to date, all classified within the species Orthohepevirus A [6]. SHEV-3 strains are predominantly prevalent in Europe, Asia, and the Americas, whereas SHEV-4 exhibits a circulation pattern primarily in Europe and Asia. In mainland China, SHEV-1 remains the predominant genotype, followed by SHEV-4 [7,8].

The swine hepatitis E virus has three overlapping open reading frames (ORFs): ORF1, ORF2, and ORF3 [9]. The SHEV open reading frame 3 (ORF3) is an important virulence protein that can encode the small multifunctional phosphoprotein [10]. Although ORF3 is the smallest open reading frame in the SHEV genome, it contains recognition sequences for various protein kinases, which are thought to play a crucial role in signal transduction and the release of virulence factors. All three ORFs of SHEV regulate many cellular signaling pathways and suppress host immune responses to promote the survival of infected cells, influencing viral replication and release, lipid metabolism, and the onset of hepatitis [11].

Circular RNA (circRNA) is a type of closed loop RNA molecule formed through back splicing. The absence of its 5′ cap and 3′ poly (A) tail, as well as its covalently closed circular structure, endow it with unique exonuclease resistance, making its stability and half-life significantly better than linear RNA in cells [12]. Although the presence of circRNA was first observed in plant viruses in the 1970s, due to technological limitations, such molecules have long been misunderstood as byproducts of transcription or splicing errors [13]. Until 2012, with the breakthrough development of high-throughput sequencing technology and bioinformatics analysis, research revealed that circRNA is widely present in eukaryotes and exhibits tissue-specific expression patterns. Its function is no longer limited to the old cognition of “molecular redundancy”—studies have shown that circRNA can finely regulate gene expression networks by adsorbing miRNA, binding to RNA binding proteins, or directly encoding small peptides, becoming a cutting-edge research direction in the field of non-coding RNA. In recent years, many studies have shown that circRNA is involved in the regulation of viral infection mechanisms, such as circVAMP3 restricting the replication of influenza A virus by interfering with NP and NS1 proteins [14], and circBACH1 promoting hepatitis B virus replication and liver cancer development by regulating the miR-200a-3p/MAP3K2 axis [15]. In our previous research, we have analyzed the circRNA expression profile in HepG2 cells overexpressing swine hepatitis E virus ORF3, but we have not discussed in detail the relationship between differentially expressed circRNA and ORF3 on the mechanism of HEV infection [16]. Therefore, in this study, based on previous circRNA and transcriptome research, we constructed a potential circRNA-miRNA-mRNA regulatory network targeting viral infection related pathways, laying the foundation for revealing the function of ORF3 in viral infection, explaining the interaction mechanism between SHEV and target cells, and providing scientific basis for the prevention and treatment of SHE.

## 2. Materials and Methods

### 2.1. Overexpression of Genotype IV SHEV pORF3 by Recombinant Adenovirus AD-ORF3 in HepG2 Cells and High-Throughput Sequencing of circRNA and Transcriptome

Gene type IV SHEV pORF3 was overexpressed in HepG2 cells by recombinant adenovirus AD-ORF3. The human hepatocellular carcinoma cell line HepG2 (purchased from the Shanghai Cell Bank of the Chinese Academy of Sciences, Shanghai, China) was routinely cultured in DMEM medium containing 10% fetal bovine serum (37 °C, 5% CO_2_). The experimental design was divided into 2 groups: recombinant adenovirus-infected group (AD-ORF3) and control virus group (AD-GFP, which carries a green fluorescent protein reporter gene). HepG2 cells were infected with the recombinant adenovirus AD-ORF3 or the control virus AD-GFP at a multiplicity of infection (MOI) of 5 for 24 h to ensure efficient gene delivery and overexpression. And high-throughput sequencing of circRNA and transcriptome was performed. Total RNA was extracted from three biological replicates per group and used for ribosomal RNA-depleted, strand-specific RNA sequencing on the Illumina HiSeq 4000 platform (LC Bio, Hangzhou, China). Sequencing reads were aligned to the Homo sapiens reference genome (Ensemble release-96), and circRNAs were identified using a combined approach with CIRCExplorer2 and CIRI under the following criteria: back-spliced junction reads ≥ 1, mismatch ≤ 2, and distance between splice sites < 100 kb. Differentially expressed circRNAs were identified using edgeR with a threshold of |log_2_(fold change)| > 1.0 and *p*-value < 0.05. Functional annotation of circRNA host genes was performed through GO and KEGG enrichment analyses. To explore potential post-transcriptional regulatory mechanisms, circRNA—miRNA interactions were predicted using TargetScan (v7.2) (context++ score percentile ≥ 50) and miRanda (v3.3a) (max free energy < −10 kcal/mol). The details and results of this method are described in our previously published article.

### 2.2. Screening for Differentially Expressed mRNAs Associated with Viral Infection Pathways

Based on gene expression profiles from six samples (Ad_GFP1, Ad_GFP2, Ad_GFP3, Ad_ORF3_1, Ad_ORF3_2, Ad_ORF3_3), differentially expressed genes associated with viral infection-related pathways were identified through KEGG functional annotation enrichment analysis, with the threshold set at |log_2_(fold change)| > 1.0 and a *p*-value < 0.05. The differential genes involved in viral infection pathways were visualized using a heatmap (Figure 1), where the horizontal axis represents the samples and the vertical axis represents the mRNAs. Distinct colors indicate different levels of mRNA expression: red denotes highly expressed mRNAs, and dark blue indicates lowly expressed mRNAs.

### 2.3. Prediction of miRNAs Associated with Differentially Expressed mRNAs of Viral Infection-Related Pathways

The screened differentially expressed mRNAs were used to predict potentially binding miRNAs through an online tool (https://jingege.shinyapps.io/jingle_molecular/ (accessed on 16 March 2025)), which integrates a total of 11 databases. These include three experimentally validated databases (miRecords, miRTarBase, and TarBase) and eight prediction databases (DIANA-microT, ElMMo, MicroCosm, miRanda, miRDB, PicTar, PITA, and TargetScan). Only miRNAs supported by five or more of these databases were selected for subsequent analysis.

### 2.4. Screening for Differentially Expressed circRNAs Associated with Viral Infection Pathways

The predicted miRNAs were subsequently used for circRNA prediction through both TargetScan (http://www.targetscan.org/ (accessed on 17 March 2025)) and miRanda (https://www.bioinformatics.com.cn/local_miranda_miRNA_target_prediction_120 (accessed on 17 March 2025)), with thresholds set as follows: a maximum free energy of < −10 kcal/mol in miRanda and a context++ score percentile ≥ 50 in TargetScan. The intersection between these prediction results and the experimentally identified differentially expressed circRNAs from sequencing data (filtered at |log_2_FC| > 1.0 and *p*-value < 0.05) was taken to identify circRNAs associated with viral infection-related pathways.

### 2.5. Prediction of circRNA-miRNA-mRNA Regulatory Network Associated with Viral Infection in Genotype IV SHEV ORF3 in HepG2 Cells

To identify relevant miRNA interactions, the screened differentially expressed circRNAs were analyzed using both TargetScan and miRanda. The resulting miRNA predictions were then intersected with the miRNA sets previously derived from mRNA-based prediction. Finally, the associations among the filtered circRNAs, miRNAs, and mRNAs were integrated and visualized as a comprehensive circRNA–miRNA–mRNA regulatory network using Cytoscape version 3.10.0.

## 3. Results

### 3.1. Screening of Differentially Expressed mRNAs of Viral Infection-Related Pathways

Differentially expressed mRNAs were screened by the following GO and KEGG functional annotations of viral infection-related pathways (Table 1 and Table 2), and a total of 32 differentially expressed mRNAs (Figure 1) were screened (Table 3).

### 3.2. Prediction of miRNAs Interacting with Differentially Expressed mRNAs Associated with Viral Infection Pathways

The 32 mRNAs screened above were used to predict their binding miRNAs through an online website tool, with 11 prediction databases including 3 experimental validation databases: mirecords, mirtarbase, tarbase, and 8 prediction databases: diana_microt, elmmo, microcosm, miranda, mirdb, pictar, pita, targetscan), a total of 6480 results were predicted, and 51 results were obtained by filtering miRNAs that could be predicted by 5 or more databases (Appendix A).

### 3.3. Screening for Differentially Expressed circRNAs Associated with Viral Infection Pathways

CircRNAs act as microRNA sponges associated with related miRNAs, which together constitute the circRNA-miRNA axis involved in disease pathogenesis. Using the 51 miRNAs obtained after post-prediction screening, circRNA prediction was performed by Targetscan and miRanda, and the intersection of the prediction results with the actual differentially expressed circRNAs in the sequencing data was taken to obtain the following 32 circRNAs (Figure 2 & Table 4).

### 3.4. Prediction of Viral Infection Pathway-Associated circRNA-miRNA-mRNA Regulatory Network

The 32 circular RNAs were screened for miRNA targeting prediction (combined with TargetScan and miRanda tools), and the miRNAs predicted by the differentially expressed mRNA related to the virus infection pathway were crossed, and finally 23 miRNAs were obtained. The obtained 23 miRNAs were associated with the corresponding 7 differentially expressed mRNAs (Appendix A), and then a potential circRNA-miRNA-mRNA regulatory network containing 32 circular RNAs, 23 miRNAs and 7 mRNAs was constructed through Cytoscape (Figure 3). Among them, a total of 167 regulatory pathways were screened (Appendix A), with 51 associated with HSPA8 (Figure 4a), 36 associated with EGR2 (Figure 4b), 28 associated with SOCS3 (Figure 4c), 22 associated with ZNF527 (Figure 4d), and CXCR4 associated with 13 (Figure 4e), 11 associated with HSPA1B (Figure 4f), and 6 associated with NOTCH3 (Figure 4g). Among them, circRNA5591, circRNA5619, ciRNA203, circRNA5562, and circRNA8848 may be shared circRNAs of multiple regulatory pathways, which appeared 98 times in 167 pathways. hsa-miR-106a-5p, hsa-miR-17-5p, hsa-miR -20a-5p, hsa-miR-93-5p, hsa-miR-34a-5p, hsa-miR-203a-3p, hsa-miR-26a-5p, and hsa-miR-26b-5p may be shared miRNAs for multiple regulatory pathways, appearing 89 times in 167 pathways.

## 4. Discussion

In this study, based on previous circRNAomics and transcriptomics sequencing data, we analyzed the changes in expression profiles of porcine hepatitis E virus ORF3 overexpression adenovirus after transfection of HepG2 cells, and firstly screened out 32 differentially expressed mRNAs related to the viral infection pathway. Subsequently, we predicted potential miRNA targets of these 32 mRNAs by using an online tool to obtain a total of 51 candidate miRNAs, and further predicted the circRNAs that these miRNAs might bind using TargetScan and miRanda software. Combined with the actual differentially expressed circRNAs in the sequencing data, 32 candidate circRNAs associated with viral regulation were screened out. Next, the potential miRNAs of these circRNAs were predicted in reverse, and compared with pre mRNA-associated miRNAs to take the intersection, and finally obtained 23 core miRNAs. Based on the above analysis, we integrated the 32 circRNAs, 23 miRNAs and the key 7 mRNAs among them, and constructed a circRNA-miRNA-mRNA regulatory network by using Cytoscape to systematically reveal the role of the hepatitis E virus ORF3 protein in viral regulation and the role of the hepatitis virus ORF3 protein in hepatocytes.

It is important to note that this study employed an adenovirus-mediated ORF3 overexpression system in HepG2 cells, a well-established model for investigating specific viral protein functions. While this approach enables precise examination of ORF3-specific effects, it presents inherent methodological considerations including non-physiological expression levels and separation from the complete viral replication context. These aspects should be considered when interpreting the identified regulatory networks.

The circRNA–miRNA–mRNA regulatory network constructed in this study encompasses seven differentially expressed mRNAs (HSPA8, HSPA1B, EGR2, SOCS3, ZNF527, CXCR4, NOTCH3). Both HSPA8 and HSPA1B belong to the heat shock protein family. Literature reports indicate that HSPA8 may interact with PRRSV GP4 protein to participate in viral attachment and internalization [17], and enhance PRRSV replication by promoting RAB18/PLIN2 interactions through chaperone-mediated autophagy [18]. HSPA8 has also been implicated in viral RNA release during JEV infection [19]. In HBV-related hepatocellular carcinoma, HSPA8 appears to exert dual functions, potentially regulating viral replication and ferroptosis pathways [20]. Similarly, HSPA1B has been shown to be required for efficient replication in ECTV infection [21], while exhibiting inhibitory effects on viral proliferation during the mid-to-late stages of ORFV infection [22], suggesting virus-specific functionalities within this protein family. Our bioinformatic predictions suggest that circRNAs such as circRNA5591, ciRNA203, and circRNA5619 may potentially influence the expression levels of HSPA8 and HSPA1B by sequestering miRNAs including hsa-miR-34a-5p and hsa-miR-26a-5p. This regulatory relationship might be associated with the progression of SHEV infection.

The early growth response transcription factor EGR2 participates in host defense through its negative regulation of T cell activation. Studies have demonstrated that EGR2 deficiency significantly reduces resistance in influenza virus-infected mice [23]. Our bioinformatic predictions suggest that circRNAs including circRNA5562, circRNA5591, and ciRNA203 might influence EGR2 expression through interactions with miRNAs such as hsa-miR-140-5p and hsa-miR-106a-5p. This potential regulatory relationship indicates that downregulation of EGR2 expression could be somewhat correlated with alterations in host defense mechanisms against SHEV.

SOCS3 functions as a negative immune regulator and demonstrates differential regulation patterns across viral infections: HSV-1 upregulates SOCS3 via STAT3 activation to suppress JAK/STAT antiviral signaling [24]; influenza A virus induces SOCS3 through an NF-κB-dependent pathway to attenuate type I interferon responses [25]; while HIV-1 downregulates SOCS3 to trigger persistent immune activation that promotes viral replication [26]. PRRSV can also promote viral replication by inducing SOCS3 expression through the p38/AP-1 signaling pathway [27]. Our predictive analysis suggests that circRNAs including circRNA5562, circRNA5619, and circRNA5591 might potentially affect SOCS3 expression by sequestering miRNAs such as hsa-miR-203a-3p and hsa-miR-19b-3p. This regulatory pattern could potentially create a favorable environment for viral infection.

Although direct evidence for ZNF527 in viral infection remains limited, zinc finger protein family members typically participate in antiviral immune responses. Research indicates that ZNF268 may contribute to antiviral immunity by promoting NF-κB signaling through maintaining IKK complex stability [28], while certain ZNF proteins might potentially inhibit SARS-CoV-2 infection by enhancing immune cell activity and abundance [29]. Our predictive results suggest that circRNAs including circRNA5119, circRNA5510, and circRNA5591 might influence ZNF527 expression by sequestering miRNAs such as hsa-miR-181c-5p and hsa-miR-181a-5p. This regulatory relationship could potentially be linked to host anti-SHEV immune responses.

The chemokine receptor CXCR4 serves as a key co-receptor for HIV-1 entry into CD4+ T cells [30]. Studies show that viruses such as HSV-1 and EBV can interfere with immune cell function by downregulating CXCR4 expression [31,32]. Our bioinformatic predictions suggest that circRNAs including circRNA17257, circRNA5591, and circRNA5619 might potentially affect CXCR4 expression by sequestering hsa-miR-139-5p and hsa-miR-204-5p. Such expression changes might be correlated with alterations in immune cell chemotactic function, potentially providing favorable conditions for SHEV infection establishment.

NOTCH3 signaling activation has been associated with various viral infection pathologies: HIV Tat protein drives neuroinflammation by enhancing NOTCH3 signaling [33]; SARS-CoV-2 infection in alveolar foci is accompanied by NOTCH3 upregulation [34]; and HCV NS3 protein promotes persistent infection by modulating the Notch pathway through SRCAP/p400 [15]. Our analysis suggests that circRNAs including ciRNA203, circRNA5199, and circRNA5510 might potentially regulate NOTCH3 expression by sequestering hsa-miR-1-3p. These predictive results imply that the NOTCH3 signaling pathway might be involved in SHEV–host cell interactions.

## 5. Conclusions

In summary, this study utilized transcriptomic and circRNA sequencing data from SHEV ORF3-overexpressing adenovirus-infected HepG2 cells to construct a circRNA-miRNA-mRNA regulatory network. Through differential expression analysis and bioinformatics prediction, the network was identified to center on seven core mRNAs (HSPA8, HSPA1B, EGR2, SOCS3, ZNF527, CXCR4, and NOTCH3). In this study, we showed that ciRNA203, circRNA14936, circRNA5510, circRNA5562, circRNA5591, circRNA5619, and circRNA8848 may act as potential key molecules in the process of SHEV infection of HepG2 cells, and through competitive binding of miRNAs that targeted regulation of the transcription of seven differentially expressed mRNAs, including HSPA8, HSPA1B, and SOCS3 (Appendix A). Although our study proposes a theoretical framework for the circRNA-miRNA-mRNA interactions of viral infection-related pathways in SHEV ORF3-overexpressing HepG2 cells, the molecular mechanism still needs to be further confirmed by target validation experiments. Moving forward, we plan to conduct both in vitro and in vivo functional experiments—such as RNA pulldown, dual-luciferase reporter assays, and gene knockdown/overexpression studies—to validate the key molecular interactions within this network. These efforts will help clarify the biological functions and molecular mechanisms of these candidates during SHEV infection, thereby contributing to a more comprehensive understanding of the virus–host interaction network.

## Figures and Tables

**Figure 1 microorganisms-13-02654-f001:**
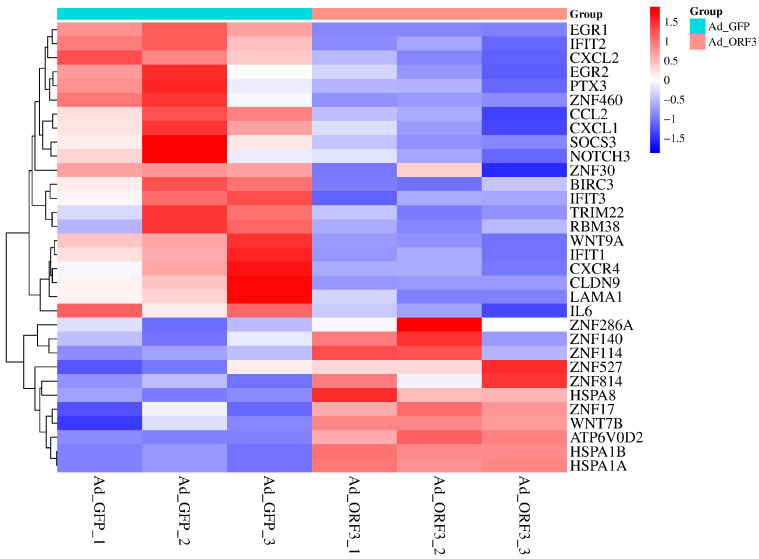
The mRNA of the pathways associated with viral infection that were selected based on transcriptome sequencing was differentially expressed.

**Figure 2 microorganisms-13-02654-f002:**
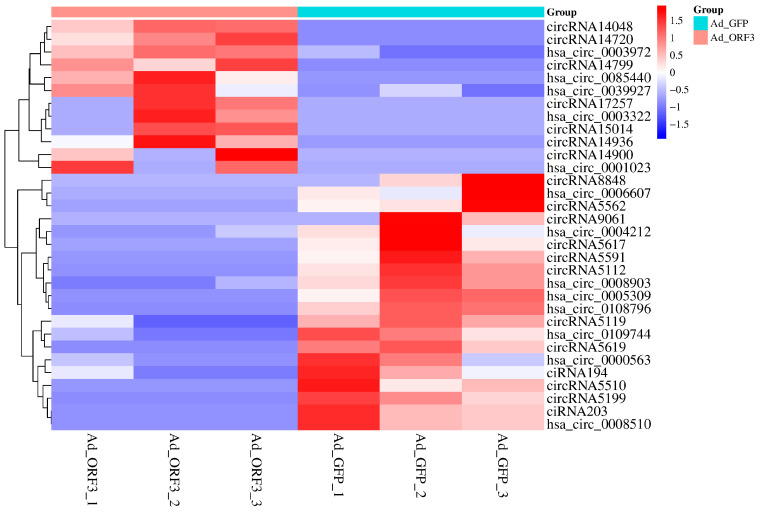
Based on circRNA sequencing, 31 differential expression circRNA related to viral infection pathways were screened out.

**Figure 3 microorganisms-13-02654-f003:**
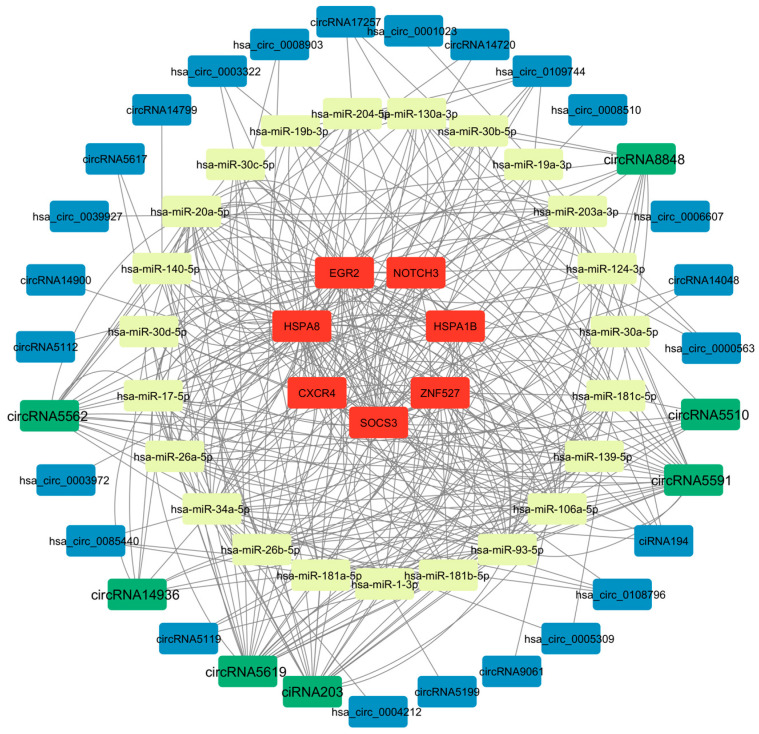
The potential circRNA-miRNA-mRNA regulatory network associated with viral infection was constructed using Cytoscape. Within this network, mRNA nodes are represented in red, miRNA nodes in yellow, and circRNA nodes in blue and green. Notably, the potential hub circRNAs identified in this study are highlighted as green nodes with enlarged font size.

**Figure 4 microorganisms-13-02654-f004:**
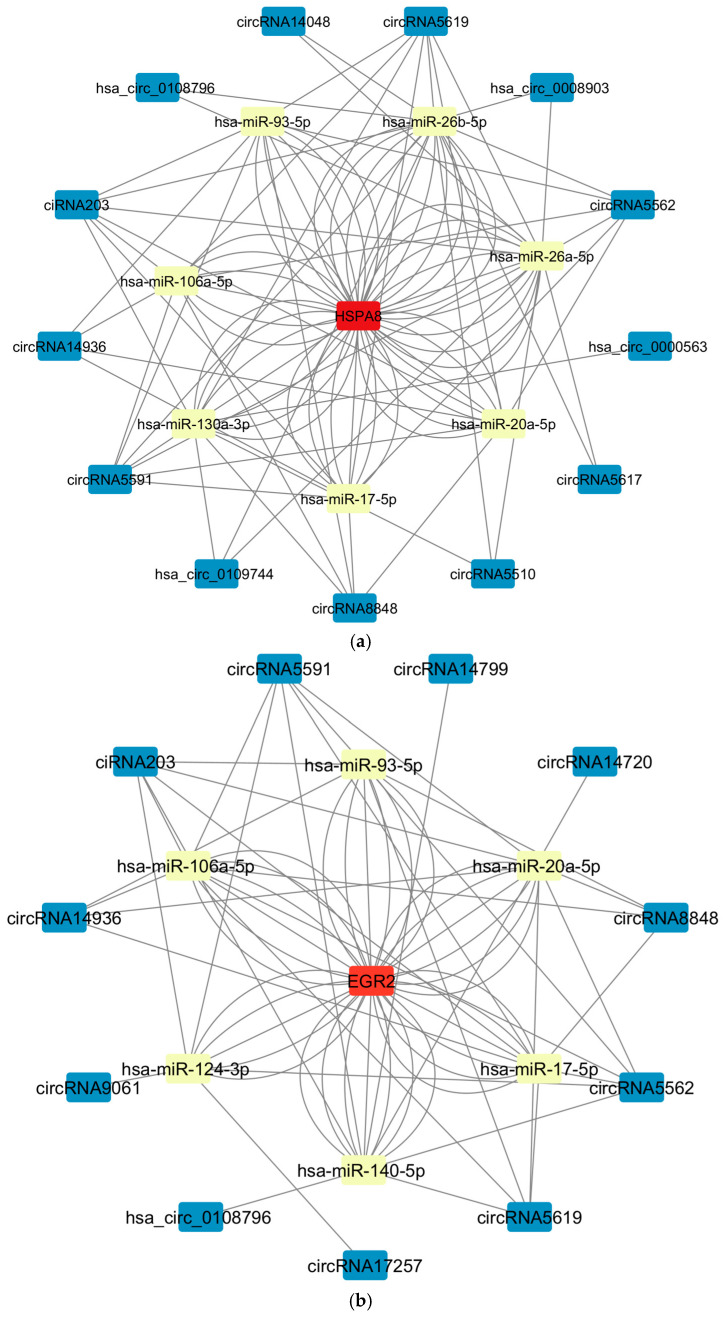
The potential circRNA-miRNA-mRNA regulatory network of each gene was constructed by Cytoscape. Within this network, mRNA nodes are represented in red, miRNA nodes in yellow, and circRNA nodes in blue. (**a**) CircRNA-miRNA-mRNA regulatory pathways of HSPA8 were demonstrated using network diagram. (**b**) CircRNA-miRNA-mRNA regulatory pathways of EGR2 were demonstrated using network diagram. (**c**) CircRNA-miRNA-mRNA regulatory pathways of SOCS3 were demonstrated using network diagram. (**d**) CircRNA-miRNA-mRNA regulatory pathways of ZNF527 were demonstrated using network diagram. (**e**) CircRNA-miRNA-mRNA regulatory pathways of CXCR4 were demonstrated using network diagram. (**f**) CircRNA-miRNA-mRNA regulatory pathways of HSPA1B were demonstrated using network diagram. (**g**) CircRNA-miRNA-mRNA regulatory pathways of NOTCH3 were demonstrated using network diagram.

**Table 1 microorganisms-13-02654-t001:** GO Pathway related to viral infection.

Pathway ID	Pathway_Name
GO:0001618	Virus receptor activity
GO:0016032	Viral process
GO:0046718	Viral entry into host cell
GO:0009615	Response to virus
GO:0045087	Innate immune response
GO:0051607	Defense response to virus
GO:0060337	Type I interferon signaling pathway
GO:0019064	Fusion of virus membrane with host membrane
GO:0045071	Negative regulation of viral genome replication
GO:0046596	Regulation of viral entry
GO:0042535	Positive regulation of TNF biosynthesis
GO:0043066	Negative regulation of apoptosis
GO:0071222	Cellular response to LPS
GO:0007165	Signal transduction
GO:0007259	JAK-STAT cascade
GO:0035458	Cellular response to interferon-beta
GO:0039528	Cytokine-mediated signaling pathway

**Table 2 microorganisms-13-02654-t002:** KEGG pathway related to viral infection.

Pathway ID	Pathway_Name
Ko04612	Antigen processing and presentation
Ko05160	Hepatitis C
Ko05161	Hepatitis B
Ko05168	Herpes simplex virus 1 infection
Ko04630	JAK-STAT signaling pathway
Ko04620	Toll-like receptor signaling pathway
Ko04622	RIG-I-like receptor signaling pathway
Ko04621	NOD-like receptor signaling pathway
Ko04658	Th1 and Th2 cell differentiation
Ko04062	Chemokine signaling pathway
Ko04668	TNF signaling pathway
Ko04145	Phagosome
Ko04217	Necroptosis
Ko04140	Autophagy
Ko04010	MAPK signaling pathway
Ko04064	NF-kappa B signaling pathway
Ko04150	mTOR signaling pathway

**Table 3 microorganisms-13-02654-t003:** The sequencing data of 32 mRNAs related to the differential expression of viral infection pathways were screened.

Gene_ID	Gene_Name	Log_2_(fc)	Regulation
ENSG00000204389	HSPA1A	2.08	up
ENSG00000204388	HSPA1B	1.86	up
ENSG00000119922	IFIT2	−3.00	down
ENSG00000121966	CXCR4	−2.06	down
ENSG00000185745	IFIT1	−1.39	down
ENSG00000213937	CLDN9	−1.56	down
ENSG00000119917	IFIT3	−1.20	down
ENSG00000136244	IL6	−2.04	down
ENSG00000132274	TRIM22	−1.35	down
ENSG00000109971	HSPA8	1.06	up
ENSG00000132819	RBM38	−1.11	down
ENSG00000163661	PTX3	−2.33	down
ENSG00000108691	CCL2	−1.99	down
ENSG00000120738	EGR1	−1.59	down
ENSG00000147614	ATP6V0D2	3.61	up
ENSG00000178150	ZNF114	1.96	up
ENSG00000081041	CXCL2	−1.94	down
ENSG00000204514	ZNF814	1.54	up
ENSG00000184557	SOCS3	−1.57	down
ENSG00000101680	LAMA1	−2.19	down
ENSG00000143816	WNT9A	−1.49	down
ENSG00000074181	NOTCH3	−1.78	down
ENSG00000187607	ZNF286A	1.25	up
ENSG00000197714	ZNF460	−1.00	down
ENSG00000023445	BIRC3	−1.13	down
ENSG00000186272	ZNF17	1.05	up
ENSG00000122877	EGR2	−1.40	down
ENSG00000188064	WNT7B	1.29	up
ENSG00000196387	ZNF140	1.38	up
ENSG00000163739	CXCL1	−1.93	down
ENSG00000189164	ZNF527	1.07	up
ENSG00000168661	ZNF30	−1.07	down

**Table 4 microorganisms-13-02654-t004:** The sequencing data of circRNA related to the differential expression of viral infection pathways were screened. (The inf values in the log_2_(fold change) column represent infinite fold changes, which occur when the expression level in one group is undetectable or below the detection limit of the assay).

circRNA	Log_2_(fc)	Regulation
circRNA5617	−inf	down
circRNA5199	−inf	down
circRNA14936	inf	up
circRNA15014	inf	up
hsa_circ_0109744	−3.34	down
circRNA5510	−inf	down
hsa_circ_0004212	−3.45	down
hsa_circ_0001023	inf	up
hsa_circ_0085440	inf	up
hsa_circ_0005309	−inf	down
circRNA5591	−inf	down
hsa_circ_0008903	−3.77	down
circRNA14799	inf	up
circRNA5562	−inf	down
circRNA14720	inf	up
circRNA9061	−inf	down
circRNA5112	−inf	down
hsa_circ_0003972	3.40	up
circRNA8848	−inf	down
circRNA14048	inf	up
circRNA5119	−2.60	down
circRNA5619	−inf	down
hsa_circ_0108796	−inf	down
ciRNA203	−inf	down
hsa_circ_0008510	−inf	down
hsa_circ_0003322	inf	up
hsa_circ_0006607	−inf	down
ciRNA194	−2.66	down
hsa_circ_0000563	−3.63	down
circRNA14900	inf	up
circRNA17257	inf	up
hsa_circ_0039927	1.02	up

## Data Availability

The original contributions presented in this study are included in the article and Appendix A. Further inquiries can be directed to the corresponding authors.

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
