# Peer review of "Comprehensive Bioinformatics Analysis the circRNAs of Viral Infection Associated Pathway in HepG2 Expressing ORF3 of Genotype IV Swine Hepatitis E Virus"

_microorganisms, 2025, doi:10.3390/microorganisms13122654_

Round 1

Reviewer 1 Report

Comments and Suggestions for Authors

1)circRNAs production could be influenced by cell lines used in the experiments. The results  should be confirmed in different cell lines and 3D systems.

2)The authors should justify their choice of databases and the software used.

3) Discussion : it is not clear the relationship between CXCR4 and HEV infection ( lines 275-280)

Author Response

Comment 1: The circRNAs production could be influenced by the cell lines used in the experiments. The results should be confirmed in different cell lines and 3D systems.

Response: We sincerely thank the reviewer for raising this critical point regarding the choice of cell model. We fully appreciate the concern that using a human hepatoma cell line (HepG2) to study a porcine virus might not perfectly mirror the natural infection in pigs. Please allow us to clarify the rationale behind our choice and how we plan to address this in our future work.

Our use of HepG2 cells is supported by their established application in SHEV research, particularly for the study of the ORF3 protein. This approach is documented in previous literature:

Firstly, in our own prior and related work, we and other researchers have successfully utilized the HepG2 model to investigate the molecular mechanisms of SHEV ORF3, including high-throughput screening of circRNA and transcriptomic profiles, demonstrating its functional utility in this context[1][2].

Secondly, and critically, genotype IV SHEV is recognized as a zoonotic pathogen with the ability to infect humans . Therefore, studying SHEV ORF3 in a human hepatoma cell line is not only methodologically feasible but also highly relevant for understanding the cross-species infection potential and the fundamental virology of this virus in a human cellular context.

We fully agree with the reviewer that confirming these findings in porcine-specific systems is a vital next step to elucidate the mechanisms in the natural host. In response to this comment, we have revised the discussion section to explicitly acknowledge this perspective and to outline our future plans for validation in porcine primary hepatocytes and/or porcine cell lines (such as PK-15 or ST cells, which are commonly used in porcine virology ) to firmly establish the physiological relevance of the identified regulatory network in swine.

  • 1.Jiao H, Zhao Y, Zhou Z, Li W, Li B, Gu G, Luo Y, Shuai X, Fan C, Wu L, Chen J, Huang Q, Wang F, Liu J. Identifying Circular RNAs in HepG2 Expressing Genotype IV Swine Hepatitis E Virus ORF3 Via Whole Genome Sequencing. Cell Transplant. 2021 Jan-Dec;30:9636897211055042. doi: 10.1177/09636897211055042
  • 2.Gong G, Xin J, Lou Y, Qiong D, Dawa Z, Gesang Z, Suolang S. Cell Culture of a Swine Genotype 4 Hepatitis E Virus Strain. J Med Virol. 2024 Nov;96(11):e70031. doi: 10.1002/jmv.70031

Comments 2:The authors should justify their choice of databases and the software used

Response:

We thank the reviewer for the opportunity to clarify the rationale behind our selection of databases and software. Our choices were guided by the principles of widespread acceptance, complementary strengths, and methodological rigor in the field of non-coding RNA bioinformatics.

  1. Justification for miRNA Prediction Databases:

The online tool (https://jingege.shinyapps.io/jingle_molecular/) was selected because it provides a comprehensive and integrated platform, aggregating results from 11 distinct databases. This approach was crucial to minimize false positives and enhance prediction reliability. Our selection criteria specifically included:

Three experimentally validated databases (miRecords, miRTarBase, TarBase): These provided high-confidence miRNA-mRNA interactions supported by empirical evidence (e.g., luciferase reporter assays, CLIP-seq).

Eight prediction databases (TargetScan, miRanda, etc.): These utilize well-established algorithms based on key biological principles, such as seed region complementarity (TargetScan), evolutionary conservation, and thermodynamic stability of miRNA-mRNA duplexes (miRanda).

By requiring that a miRNA-mRNA pair be supported by at least 5 of these 11 databases, we enforced a stringent consensus, ensuring that only the most robust interactions were carried forward for network construction.

  1. Justification for circRNA-miRNA Prediction Software:

For predicting circRNA-miRNA interactions, we employed a dual-tool strategy using TargetScan and miRanda.

TargetScan is a gold-standard tool renowned for its focus on conserved seed matches and its context-specific prediction model.

miRanda complements this by evaluating sequence complementarity and calculating the thermodynamic stability (free energy) of the potential RNA duplex.

We applied stringent thresholds for both tools (TargetScan score percentile ≥ 50; miRanda max free energy < -10) to select for high-affinity, biologically plausible interactions. Using the intersection of predictions from these two independent and highly cited algorithms significantly increases the confidence in our resulting circRNA-miRNA-mRNA network.

We hope this clarification satisfactorily addresses the reviewer's question regarding our bioinformatic methodology.

Comment 3: Discussion: it is not clear the relationship between CXCR4 and HEV infection (lines 275-280).

Response: We thank the reviewer for this critical comment. We agree that the relationship between CXCR4 and HEV infection is not yet established, and we have toned down our discussion to reflect the speculative nature of our bioinformatics-based prediction.

Our analysis solely indicates that CXCR4 is a downstream target within the predicted circRNA-miRNA-mRNA network. The revised discussion now explicitly states that this finding should be considered a hypothesis-generating observation rather than evidence of a functional link. We have rephrased the text to propose that the downregulation of CXCR4 could represent a potential mechanism, drawing a parallel to strategies employed by other viruses, while clearly emphasizing that this idea requires direct experimental validation in future studies.

Reviewer 2 Report

Comments and Suggestions for Authors

The manuscript addresses an interesting and relevant question: how the SHEV ORF3 protein affects host cell regulatory networks, focusing on circRNA–miRNA–mRNA interactions. The integration of circRNAomics and transcriptomics is scientifically sound and highly valuable, given the scarcity of mechanistic studies on SHEV. The paper contains rich datasets and a thoughtful discussion about potential mechanisms, including involvement of HSPA8/HSPA1B, SOCS3, EGR2, ZNF family proteins, and CXCR4. However, the manuscript currently suffers from several issues that may weaken its impact if not addressed.

Major points

  1. The abstract is long and dense, with several grammatical inconsistencies and ambiguous phrasing. It should be reorganized to state: Background, Objective, Methods, Conclusions clearly
  2. Methodological description is insufficient.
  3. The Discussion repeatedly cites: “previous sequencing”, “screened X circRNAs”, and prediction methodology (TargetScan, miRanda). But key parameters are missing: fold-change cutoffs, p-value / FDR thresholds, software versions, reference genomes used, filtering criteria (e.g., circRNA detection thresholds), adenovirus multiplicity of infection (MOI), duration of expression

These should ideally be in Methods, but the Discussion references them extensively and should not assume the reader has access to details elsewhere.

  1. Some paragraphs lose coherence by mixing: previous literature, your predictions, and speculative roles of circRNAs

non-coding RNA theory, downstream effects

Try structuring each gene section as: Known role of the gene in viral infection, Your findings (circRNA–miRNA–mRNA links), Biological implications relevant to SHEV

  1. The Discussion frequently uses strong causal language (e.g., “may enhance viral replication”) even though the study is purely predictive. You acknowledge limitations later, but it is crucial to adjust phrasing earlier to reflect association and hypothesis generation, not demonstration.
  2. Multiple sections list long sets of circRNAs/miRNAs that overwhelm the narrative. Instead: move complete lists to Supplementary Tables; summarize in text (“six circRNAs including circRNA5562 and circRNA5619…”)
  3. Clarification about the adenoviral overexpression model. The added justification is appreciated and scientifically valid. However, It should appear in the Discussion, not at the end of the manuscript. It needs a more neutral tone and focus on methodological rigor rather than defending the method.
  4. Revision for English clarity is necessary (typographical and grammatical errors).

Minor points

  1. ZNF numbering inconsistency; ZNF257 vs ZNF527 appears. Confirm which is correct.
  2. miRNA notation; Standard form is miR-xxx-5p without “hsa-” unless cross-species relevance is crucial.
  3. Some statements are oversimplified (e.g., “ZNF proteins inhibit SARS-CoV-2”), which should be rephrased more cautiously.
  4. Multiple paragraphs repeat the same circRNAs; this should be condensed.
  5. Separate discussion for conclusions as an independent item.
  6. Tables 3, 4, 5, 6, 7, and 8 should be presented as supplementary.

Comments on the Quality of English Language

A revision for English clarity is necessary (typographical and grammatical errors).

Author Response

Comment 1:The abstract is long and dense, with several grammatical inconsistencies and ambiguous phrasing. It should be reorganized to state: Background, Objective, Methods, Conclusions clearly

Response :

We sincerely thank the reviewer for this constructive feedback. We agree that the abstract could be improved for better clarity and structure. As suggested, we have thoroughly revised the abstract to clearly delineate the Background, Objective, Methods, and Conclusions. We have also carefully addressed the grammatical inconsistencies and ambiguous phrasing to enhance readability and precision.

Comment 2:Methodological description is insufficient.

Response :

We sincerely thank the reviewer for this critical comment regarding the methodological description. We agree that providing comprehensive methodological details is crucial for the reproducibility of our study.In response, we have thoroughly revised the Materials and Methods section to include all the key parameters and details that were previously missing.

Comment 3:The Discussion repeatedly cites: “previous sequencing”, “screened X circRNAs”, and prediction methodology (TargetScan, miRanda). But key parameters are missing: fold-change cutoffs, p-value / FDR thresholds, software versions, reference genomes used, filtering criteria (e.g., circRNA detection thresholds), adenovirus multiplicity of infection (MOI), duration of expression

Response :

We sincerely thank the reviewer for this critical observation. We agree that stating the key methodological parameters is essential for interpreting the findings discussed in the manuscript.

In direct response to this comment, we have now comprehensively revised the Materials and Methods section to include all the missing details. Specifically, we have added:

The adenovirus infection parameters: MOI of 5 and duration of 24 hours.The statistical thresholds for differential expression: |log2(fold change)| > 1.0 and p-value < 0.05.The reference genome used for alignment: Homo sapiens (Ensembl release-96).The circRNA detection criteria: back-spliced junction (BS) reads ≥ 1, mismatch ≤ 2, and distance between splice sites < 100 kb.

The versions of the bioinformatics software used, including TargetScan (v7.2), miRanda (v3.3a).

We believe that by integrating these essential parameters into the methodology, we have significantly improved the clarity, reproducibility, and overall strength of our study. We are grateful for the reviewer's insightful suggestion.

Comment 4:Some paragraphs lose coherence by mixing: previous literature, your predictions, and speculative roles of circRNAs non-coding RNA theory, downstream effects

Try structuring each gene section as: Known role of the gene in viral infection, Your findings (circRNA–miRNA–mRNA links), Biological implications relevant to SHEV

Response:

We sincerely thank the reviewer for this insightful critique regarding the logical flow of our Discussion. We agree that the previous version lacked clear structure, which weakened the argument. In response, we have comprehensively reorganized the Discussion by adopting a clear and consistent narrative framework for each gene. Specifically, for every key mRNA (e.g., HSPA8, CXCR4, SOCS3), we now first establish its well-documented role in viral infection based on existing literature; we then cleanly present our specific bioinformatic predictions of the circRNA-miRNA-mRNA interactions; finally, we build upon this foundation to propose a focused and coherent hypothesis about the potential biological implications and mechanisms of this regulatory axis specifically in the context of SHEV ORF3-mediated infection. This restructuring has significantly enhanced the clarity, logical coherence, and scholarly rigor of our interpretation by distinctly separating established knowledge, our own findings, and subsequent speculation. We are grateful for this valuable suggestion.

Comment 5:The Discussion frequently uses strong causal language (e.g., “may enhance viral replication”) even though the study is purely predictive. You acknowledge limitations later, but it is crucial to adjust phrasing earlier to reflect association and hypothesis generation, not demonstration.

Response:

We thank the reviewer for this critical observation. We completely agree that the language used in the Discussion must accurately reflect the predictive and hypothesis-generating nature of our bioinformatic study. In response, we have meticulously gone through the entire Discussion section and replaced strong causal language with phrasing that emphasizes association, prediction, and potentiality.

Comment 6:Multiple sections list long sets of circRNAs/miRNAs that overwhelm the narrative. Instead: move complete lists to Supplementary Tables; summarize in text (“six circRNAs including circRNA5562 and circRNA5619…”)

Response:We thank the reviewer for this constructive suggestion. In response, we have relocated the complete lists of circRNAs, miRNAs, and their interactions from the main text to the Supplementary Materials (Tables 4, 6, 7, and 8). The main text now summarizes these findings concisely, using phrases such as "six circRNAs, including circRNA5562 and circRNA5619..." or "*a subset of miRNAs (e.g., hsa-miR-139-5p and hsa-miR-204-5p)...*" where appropriate. This adjustment improves the narrative flow of the manuscript while ensuring full data accessibility for interested readers.

Comment 7:Clarification about the adenoviral overexpression model. The added justification is appreciated and scientifically valid. However, It should appear in the Discussion, not at the end of the manuscript. It needs a more neutral tone and focus on methodological rigor rather than defending the method.

Response:

We thank the reviewer for this valuable suggestion regarding the placement and tone of the justification for the adenoviral overexpression model. We have followed the recommendation and have now moved this discussion to the Discussion section of the manuscript.

Furthermore, we have carefully revised the text to adopt a more neutral and objective tone, focusing on the methodological rationale and the established use of this model in the field, rather than presenting it as a defensive justification. The revised text now simply states the scientific basis for the model's use and explicitly acknowledges its inherent limitations as a standard part of rigorous scientific reporting.

Comments 8:Revision for English clarity is necessary (typographical and grammatical errors).

Response:We thank the reviewer for highlighting the need for language improvement. We have thoroughly revised the entire manuscript to enhance its English clarity, focusing on correcting typographical and grammatical errors, improving sentence structure, and ensuring precise scientific terminology. This revision was conducted with careful attention to detail to present our findings in clear and accurate academic English.

Comment 8:

Response:

We sincerely thank the reviewer for pointing out the inconsistency in the zinc finger protein gene nomenclature. Upon careful verification, we confirm that ZNF527 is the correct gene symbol referenced in our study. The appearance of 'ZNF257' was an error that occurred during the analysis and writing phase. We have thoroughly corrected this throughout the entire manuscript, including the main figures, tables, and the text, to ensure that all references are unified to ZNF527. We appreciate the reviewer's meticulous attention to detail, which has been crucial in enhancing the accuracy of our work.

Comment 9:miRNA notation; Standard form is miR-xxx-5p without “hsa-” unless cross-species relevance is crucial.

Response:We thank the reviewer for raising the point regarding miRNA nomenclature. We have given this careful consideration and have decided to retain the species prefix "hsa-" for the miRNAs identified in our study.

This decision is based on the following scientific rationale: Swine Hepatitis E Virus (SHEV), particularly genotype 4 as investigated here, is a recognized zoonotic pathogen capable of crossing species barriers and infecting humans. Our study was conducted specifically in human HepG2 cells, and the identified circRNA-miRNA-mRNA network is therefore of human origin. Retaining the "hsa-" prefix is critical to:Accurately reflect the human cellular context of our experimental system.Emphasize the potential relevance of our findings to human health and zoonotic infection.Avoid potential confusion about the species origin of the regulatory molecules in subsequent studies or database searches.

We believe that maintaining this level of specificity is scientifically necessary and enhances the clarity and precision of our work in the context of a zoonotic virus.

Comment 10:Some statements are oversimplified (e.g., “ZNF proteins inhibit SARS-CoV-2”), which should be rephrased more cautiously.

Reponse:

We thank the reviewer for this pertinent observation. We agree that the statement regarding ZNF proteins and SARS-CoV-2 was overly simplistic. We have revised the text to more accurately reflect the speculative nature of this association, based on the cited literature.

Comment 11:Multiple paragraphs repeat the same circRNAs; this should be condensed.

Response:

We thank the reviewer for this valuable suggestion. We have thoroughly revised the manuscript to condense the repetitive listings of the same circRNAs across multiple paragraphs. In both the Results and Discussion sections, we have consolidated these references by using summary statements that refer to the core group of circRNAs (e.g., "key circRNAs including ciRNA203, circRNA14936...") and consistently directing readers to the complete lists in the Supplementary Tables for details. This revision significantly improves the conciseness and readability of the text.

Comment 12:Separate discussion for conclusions as an independent item.

Response :We thank the reviewer for this valuable suggestion. We have thoroughly revised the manuscript to condense the repetitive listings of the same circRNAs across multiple paragraphs. The repetitive enumerations of circRNAs (e.g., ciRNA203, circRNA5562, circRNA5591, etc.) have been consolidated. In both the Results and Discussion sections, we now refer to this core group collectively (e.g., "key circRNAs including ciRNA203, circRNA5562...") after their initial detailed introduction, and consistently direct readers to the relevant Supplementary Tables for the complete and specific interaction lists. This revision has significantly improved the conciseness, flow, and overall readability of the text.

Comment 13:Tables 3, 4, 5, 6, 7, and 8 should be presented as supplementary.

Response :

We thank the reviewer for this suggestion regarding the presentation of tables. In response, we have moved Tables 4, 6, 7, and 8 to the Supplementary Materials as recommended. However, we have retained Tables 3 and 5 in the main text because they present the core datasets of differentially expressed mRNAs and circRNAs identified through our sequencing analysis. We believe that keeping these fundamental results in the main text allows readers to better appreciate the primary findings that form the basis for our subsequent bioinformatic predictions and network construction.

Round 2

Reviewer 2 Report

Comments and Suggestions for Authors

The authors have thoroughly revised the manuscript and incorporated all suggested modifications. We now regard the manuscript as ready for publication.